# Continuous Adaptation via Meta-Learning in Nonstationary and Competitive Environments

**Maruan Al-Shedivat**[*]
CMU

**Trapit Bansal**
UMass Amherst

**Yura Burda**
OpenAI

**Ilya Sutskever**
OpenAI

**Igor Mordatch**
OpenAI

**Pieter Abbeel**
UC Berkeley

## Abstract

The ability to continuously learn and adapt from limited experience in nonstationary environments is an important milestone on the path towards general intelligence. In this paper, we cast the problem of continuous adaptation into the learning-to-learn framework. We develop a simple gradient-based meta-learning algorithm suitable for adaptation in dynamically changing and adversarial scenarios. Additionally, we design a new multi-agent competitive environment, `RoboSumo`, and define *iterated adaptation games* for testing various aspects of continuous adaptation. We demonstrate that meta-learning enables significantly more efficient adaptation than reactive baselines in the few-shot regime. Our experiments with a population of agents that learn and compete suggest that meta-learners are the fittest.

## 1 Introduction

Recent progress in reinforcement learning (RL) has achieved very impressive results ranging from playing games (Mnih et al., 2015; Silver et al., 2016), to applications in dialogue systems (Li et al., 2016), to robotics (Levine et al., 2016). Despite the progress, the learning algorithms for solving many of these tasks are designed to deal with stationary environments. On the other hand, real-world is often nonstationary either due to complexity (Sutton et al., 2007), changes in the dynamics or the objectives in the environment over the life-time of a system (Thrun, 1998), or presence of multiple learning actors (Lowe et al., 2017; Foerster et al., 2017a). Nonstationarity breaks the standard assumptions and requires agents to continuously adapt, both at training and execution time, in order to succeed.

Learning under nonstationary conditions is challenging. The classical approaches to dealing with nonstationarity are usually based on context detection (Da Silva et al., 2006) and tracking (Sutton et al., 2007), i.e., reacting to the already happened changes in the environment by continuously fine-tuning the policy. Unfortunately, modern deep RL algorithms, while able to achieve super-human performance on certain tasks, are known to be sample inefficient. Nevertheless, nonstationarity allows only for limited interaction before the properties of the environment change. Thus, it immediately puts learning into the few-shot regime and often renders simple fine-tuning methods impractical.

A nonstationary environment can be seen as a sequence of stationary tasks, and hence we propose to tackle it as a multi-task learning problem (Caruana, 1998). The learning-to-learn (or meta-learning) approaches (Schmidhuber, 1987; Thrun & Pratt, 1998) are particularly appealing in the few-shot regime, as they produce flexible learning rules that can generalize from only a handful of examples. Meta-learning has shown promising results in the supervised domain and have gained a lot of attention from the research community recently (e.g., Santoro et al., 2016; Ravi & Larochelle, 2016). In this paper, we develop a gradient-based meta-learning algorithm similar to (Finn et al., 2017b) and suitable for continuous adaptation of RL agents in nonstationary environments. More concretely, our agents meta-learn to anticipate the changes in the environment and update their policies accordingly.

While virtually any changes in an environment could induce nonstationarity (e.g., changes in the physics or characteristics of the agent), environments with multiple agents are particularly challenging

---

[*]Correspondence: `maruan.alshedivat.com`. Work done while MA and TB interned at OpenAI.

due to complexity of the emergent behavior and are of practical interest with applications ranging from multiplayer games (Peng et al., 2017) to coordinating self-driving fleets Cao et al. (2013). Multi-agent environments are nonstationary from the perspective of any individual agent since all actors are learning and changing concurrently (Lowe et al., 2017). In this paper, we consider the problem of *continuous adaptation to a learning opponent* in a competitive multi-agent setting.

To this end, we design `RoboSumo`—a 3D environment with simulated physics that allows pairs of agents to compete against each other. To test continuous adaptation, we introduce *iterated adaptation games*—a new setting where a trained agent competes against the same opponent for multiple rounds of a repeated game, while both are allowed to update their policies and change their behaviors between the rounds. In such iterated games, from the agent's perspective, the environment changes from round to round, and the agent ought to adapt in order to win the game. Additionally, the competitive component of the environment makes it not only nonstationary but also adversarial, which provides a natural training curriculum and encourages learning robust strategies (Bansal et al., 2018).

We evaluate our meta-learning agents along with a number of baselines on a (single-agent) locomotion task with handcrafted nonstationarity and on iterated adaptation games in `RoboSumo`. Our results demonstrate that meta-learned strategies clearly dominate other adaptation methods in the few-shot regime in both single- and multi-agent settings. Finally, we carry out a large-scale experiment where we train a diverse population of agents with different morphologies, policy architectures, and adaptation methods, and make them interact by competing against each other in iterated games. We evaluate the agents based on their TrueSkills (Herbrich et al., 2007) in these games, as well as evolve the population as whole for a few generations—the agents that lose disappear, while the winners get duplicated. Our results suggest that the agents with meta-learned adaptation strategies end up being the fittest. Videos that demonstrate adaptation behaviors are available at `https://goo.gl/tboqaN`.

## 2 RELATED WORK

The problem of *continuous adaptation* considered in this work is a variant of *continual learning* (Ring, 1994; 1997) and is related to *lifelong* (Thrun & Pratt, 1998; Silver et al., 2013) and *never-ending* (Mitchell et al., 2015) learning. Life-long learning systems aim at solving multiple tasks sequentially by efficiently transferring and utilizing knowledge from already learned tasks to new tasks while minimizing the effect of catastrophic forgetting (McCloskey & Cohen, 1989). Never-ending learning is concerned with mastering a fixed set of tasks in iterations, where the set keeps growing and the performance on all the tasks in the set keeps improving from iteration to iteration.

The scope of continuous adaptation is narrower and more precise. While life-long and never-ending learning settings are defined as general multi-task problems (Silver et al., 2013; Mitchell et al., 2015), continuous adaptation targets to solve a single but nonstationary task or environment. The nonstationarity in the former two problems exists and is dictated by the selected sequence of tasks. In the latter case, we assume that nonstationarity is caused by some underlying dynamics in the properties of a given task in the first place (e.g., changes in the behavior of other agents in a multi-agent setting). Finally, in the life-long and never-ending scenarios the boundary between training and execution is blurred as such systems constantly operate in the training regime. Continuous adaptation, on the other hand, expects a (potentially trained) agent to adapt to the changes in the environment at execution time under the pressure of limited data or interaction experience between the changes[1].

Nonstationarity of multi-agent environments is a well known issue that has been extensively studied in the context of learning in simple multi-player iterated games (such as rock-paper-scissors) where each episode is one-shot interaction (Singh et al., 2000; Bowling, 2005; Conitzer & Sandholm, 2007). In such games, discovering and converging to a Nash equilibrium strategy is a success for the learning agents. Modeling and exploiting opponents (Zhang & Lesser, 2010; Mealing & Shapiro, 2013) or even their learning processes (Foerster et al., 2017b) is advantageous as it improves convergence or helps to discover equilibria of certain properties (e.g., leads to cooperative behavior). In contrast, each episode in `RoboSumo` consists of multiple steps, happens in continuous time, and requires learning a good intra-episodic controller. Finding Nash equilibria in such a setting is hard. Thus, fast adaptation becomes one of the few viable strategies against changing opponents.

---

[1]The limited interaction aspect of continuous adaptation makes the problem somewhat similar to the recently proposed life-long few-shot learning (Finn et al., 2017a).

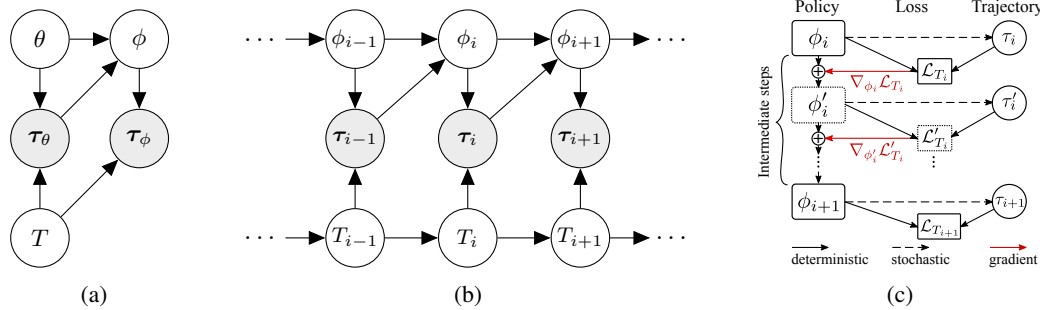

Fig. 1: (a) A probabilistic model for MAML in a multi-task RL setting. The task, $T$, the policies, $\pi$, and the trajectories, $\tau$, are all random variables with dependencies encoded in the edges of the given graph. (b) Our extended model suitable for continuous adaptation to a task changing dynamically due to non-stationarity of the environment. Policy and trajectories at a previous step are used to construct a new policy for the current step. (c) Computation graph for the meta-update from $\phi_i$ to $\phi_{i+1}$. Boxes represent replicas of the policy graphs with the specified parameters. The model is optimized via truncated backpropagation through time starting from $\mathcal{L}_{T_{i+1}}$.

Our proposed method for continuous adaptation follows the general meta-learning paradigm (Schmidhuber, 1987; Thrun & Pratt, 1998), i.e., it learns a high-level procedure that can be used to generate a good policy each time the environment changes. There is a wealth of work on meta-learning, including methods for learning update rules for neural models that were explored in the past (Bengio et al., 1990; 1992; Schmidhuber, 1992), and more recent approaches that focused on learning optimizers for deep networks (Hochreiter et al., 2001; Andrychowicz et al., 2016; Li & Malik, 2016; Ravi & Larochelle, 2016), generating model parameters (Ha et al., 2016; Edwards & Storkey, 2016; Al-Shedivat et al., 2017), learning task embeddings (Vinyals et al., 2016; Snell et al., 2017) including memory-based approaches (Santoro et al., 2016), learning to learn implicitly via RL (Wang et al., 2016; Duan et al., 2016), or simply learning a good initialization (Finn et al., 2017b).

## 3 METHOD

The problem of continuous adaptation in nonstationary environments immediately puts learning into the few-shot regime: the agent must learn from only limited amount of experience that it can collect before its environment changes. Therefore, we build our method upon the previous work on gradient-based model-agnostic meta-learning (MAML) that has been shown successful in the few-shot settings (Finn et al., 2017b). In this section, we re-derive MAML for multi-task reinforcement learning from a probabilistic perspective (*cf.* Grant et al., 2018), and then extend it to dynamically changing tasks.

### 3.1 A PROBABILISTIC VIEW OF MODEL-AGNOSTIC META-LEARNING (MAML)

Assume that we are given a distribution over tasks, $\mathcal{D}(T)$, where each task, $T$, is a tuple:

$$T := (L_T, P_T(\mathbf{x}), P_T(\mathbf{x}_{t+1} \mid \mathbf{x}_t, \mathbf{a}_t), H) \qquad (1)$$

$L_T$ is a task-specific loss function that maps a trajectory, $\tau := (\mathbf{x}_0, \mathbf{a}_1, \mathbf{x}_1, R_1, \ldots, \mathbf{a}_H, \mathbf{x}_H, R_H) \in \mathcal{T}$, to a loss value, i.e., $L_T : \mathcal{T} \mapsto \mathbb{R}$; $P_T(\mathbf{x})$ and $P_T(\mathbf{x}_{t+1} \mid \mathbf{x}_t, \mathbf{a}_t)$ define the Markovian dynamics of the environment in task $T$; $H$ denotes the horizon; observations, $\mathbf{x}_t$, and actions, $\mathbf{a}_t$, are elements (typically, vectors) of the observation space, $\mathcal{X}$, and action space, $\mathcal{A}$, respectively. The loss of a trajectory, $\tau$, is the negative cumulative reward, $L_T(\tau) := -\sum_{t=1}^{H} R_t$.

The goal of meta-learning is to find a procedure which, given access to a limited experience on a task sampled from $\mathcal{D}(T)$, can produce a good policy for solving it. More formally, after querying $K$ trajectories from a task $T \sim \mathcal{D}(T)$ under policy $\pi_\theta$, denoted $\tau_\theta^{1:K}$, we would like to construct a new, task-specific policy, $\pi_\phi$, that would minimize the expected subsequent loss on the task $T$. In particular, MAML constructs parameters of the task-specific policy, $\phi$, using gradient of $L_T$ w.r.t. $\theta$:

$$\phi := \theta - \alpha \nabla_\theta L_T\left(\tau_\theta^{1:K}\right), \text{ where } L_T\left(\tau_\theta^{1:K}\right) := \frac{1}{K}\sum_{k=1}^{K} L_T(\tau_\theta^k), \text{ and } \tau_\theta^k \sim P_T(\tau \mid \theta) \qquad (2)$$

---

**Algorithm 1** Meta-learning at training time.

**input** Distribution over pairs of tasks, $\mathcal{P}(T_i, T_{i+1})$, learning rate, $\beta$.
1: Randomly initialize $\theta$ and $\alpha$.
2: **repeat**
3:     Sample a batch of task pairs, $\{(T_i, T_{i+1})\}_{i=1}^n$.
4:     **for all** task pairs $(T_i, T_{i+1})$ in the batch **do**
5:         Sample traj. $\boldsymbol{\tau}_\theta^{1:K}$ from $T_i$ using $\pi_\theta$.
6:         Compute $\phi = \phi(\boldsymbol{\tau}_\theta^{1:K}, \theta, \alpha)$ as given in (7).
7:         Sample traj. $\boldsymbol{\tau}_\phi$ from $T_{i+1}$ using $\pi_\phi$.
8:     **end for**
9:     Compute $\nabla_\theta \mathcal{L}_{T_i, T_{i+1}}$ and $\nabla_\alpha \mathcal{L}_{T_i, T_{i+1}}$ using $\boldsymbol{\tau}_\theta^{1:K}$ and $\boldsymbol{\tau}_\phi$ as given in (8).
10:     Update $\theta \leftarrow \theta + \beta \nabla_\theta \mathcal{L}_T(\theta, \alpha)$.
11:     Update $\alpha \leftarrow \alpha + \beta \nabla_\alpha \mathcal{L}_T(\theta, \alpha)$.
12: **until** Convergence
**output** Optimal $\theta^*$ and $\alpha^*$.

---

**Algorithm 2** Adaptation at execution time.

**input** A stream of tasks, $T_1, T_2, T_3, \ldots$.
1: Initialize $\phi = \theta$.
2: **while** there are new incoming tasks **do**
3:     Get a new task, $T_i$, from the stream.
4:     Solve $T_i$ using $\pi_\phi$ policy.
5:     While solving $T_i$, collect trajectories, $\boldsymbol{\tau}_{i,\phi}^{1:K}$.
6:     Update $\phi \leftarrow \phi(\boldsymbol{\tau}_{i,\phi}^{1:K}, \theta^*, \alpha^*)$ using importance-corrected meta-update as in (9).
7: **end while**

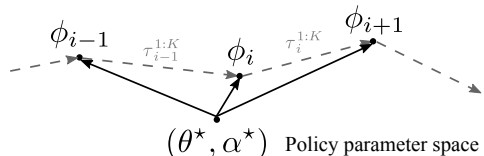

$(\theta^\star, \alpha^\star)$    Policy parameter space

---

We call (2) the *adaptation update* with a step $\alpha$. The adaptation update is parametrized by $\theta$, which we optimize by minimizing the expected loss over the distribution of tasks, $\mathcal{D}(T)$—the *meta-loss*:

$$\min_\theta \mathbb{E}_{T \sim \mathcal{D}(T)} \left[ \mathcal{L}_T(\theta) \right], \text{ where } \mathcal{L}_T(\theta) := \mathbb{E}_{\boldsymbol{\tau}_\theta^{1:K} \sim P_T(\boldsymbol{\tau}|\theta)} \left[ \mathbb{E}_{\boldsymbol{\tau}_\phi \sim P_T(\boldsymbol{\tau}|\phi)} \left[ L_T(\boldsymbol{\tau}_\phi) \mid \boldsymbol{\tau}_\theta^{1:K}, \theta \right] \right] \quad (3)$$

where $\boldsymbol{\tau}_\theta$ and $\boldsymbol{\tau}_\phi$ are trajectories obtained under $\pi_\theta$ and $\pi_\phi$, respectively.

In general, we can think of the task, trajectories, and policies, as random variables (Fig. 1a), where $\phi$ is generated from some conditional distribution $P_T(\phi \mid \theta, \boldsymbol{\tau}_{1:k})$. The meta-update (2) is equivalent to assuming the delta distribution, $P_T(\phi \mid \theta, \boldsymbol{\tau}_{1:k}) := \delta \left( \theta - \alpha \nabla_\theta \frac{1}{K} \sum_{k=1}^K L_T(\boldsymbol{\tau}_k) \right)^2$. To optimize (3), we can use the policy gradient method (Williams, 1992), where the gradient of $\mathcal{L}_T$ is as follows:

$$\nabla_\theta \mathcal{L}_T(\theta) = \mathbb{E}_{\substack{\boldsymbol{\tau}_\theta^{1:K} \sim P_T(\boldsymbol{\tau}|\theta) \\ \boldsymbol{\tau}_\phi \sim P_T(\boldsymbol{\tau}|\phi)}} \left[ L_T(\boldsymbol{\tau}_\phi) \left[ \nabla_\theta \log \pi_\phi(\boldsymbol{\tau}_\phi) + \nabla_\theta \sum_{k=1}^K \log \pi_\theta(\boldsymbol{\tau}_\theta^k) \right] \right] \quad (4)$$

The expected loss on a task, $\mathcal{L}_T$, can be optimized with trust-region policy (TRPO) (Schulman et al., 2015a) or proximal policy (PPO) (Schulman et al., 2017) optimization methods. For details and derivations please refer to Appendix A.

## 3.2 Continuous adaptation via meta-learning

In the classical multi-task setting, we make no assumptions about the distribution of tasks, $\mathcal{D}(T)$. When the environment is nonstationary, we can see it as a sequence of stationary tasks on a certain timescale where the tasks correspond to different dynamics of the environment. Then, $\mathcal{D}(T)$ is defined by the environment changes, and the tasks become sequentially dependent. Hence, we would like to exploit this dependence between consecutive tasks and meta-learn a rule that keeps updating the policy in a way that minimizes the total expected loss encountered during the interaction with the changing environment. For instance, in the multi-agent setting, when playing against an opponent that changes its strategy incrementally (e.g., due to learning), our agent should ideally meta-learn to anticipate the changes and update its policy accordingly.

In the probabilistic language, our nonstationary environment is equivalent to a distribution of tasks represented by a Markov chain (Fig. 1b). The goal is to minimize the expected loss over the chain of tasks of some length $L$:

$$\min_\theta \mathbb{E}_{\mathcal{P}(T_0), \mathcal{P}(T_{i+1}|T_i)} \left[ \sum_{i=1}^L \mathcal{L}_{T_i, T_{i+1}}(\theta) \right] \quad (5)$$

---

[2]Grant et al. (2018) similarly reinterpret adaptation updates (in non-RL settings) as Bayesian inference.

Here, $\mathcal{P}(T_0)$ and $\mathcal{P}(T_{i+1} \mid T_i)$ denote the initial and the transition probabilities in the Markov chain of tasks. Note that (i) we deal with Markovian dynamics on two levels of hierarchy, where the upper level is the dynamics of the tasks and the lower level is the MDPs that represent particular tasks, and (ii) the objectives, $\mathcal{L}_{T_i, T_{i+1}}$, will depend on the way the meta-learning process is defined. Since we are interested in adaptation updates that are optimal with respect to the Markovian transitions between the tasks, we define the meta-loss on a *pair of consecutive tasks* as follows:

$$\mathcal{L}_{T_i, T_{i+1}}(\theta) := \mathbb{E}_{\boldsymbol{\tau}_{i,\theta}^{1:K} \sim P_{T_i}(\boldsymbol{\tau}|\theta)} \left[ \mathbb{E}_{\boldsymbol{\tau}_{i+1,\phi} \sim P_{T_{i+1}}(\boldsymbol{\tau}|\phi)} \left[ L_{T_{i+1}}(\boldsymbol{\tau}_{i+1,\phi}) \mid \boldsymbol{\tau}_{i,\theta}^{1:K}, \theta \right] \right] \tag{6}$$

The principal difference between the loss in (3) and (6) is that trajectories $\boldsymbol{\tau}_{i,\theta}^{1:K}$ come from the current task, $T_i$, and are used to construct a policy, $\pi_\phi$, that is good for the upcoming task, $T_{i+1}$. Note that even though the policy parameters, $\phi_i$, are sequentially dependent (Fig. 1b), in (6) we always start from the initial parameters, $\theta$ [3]. Hence, optimizing $\mathcal{L}_{T_i, T_{i+1}}(\theta)$ is equivalent to truncated backpropagation through time with a unit lag in the chain of tasks.

To construct parameters of the policy for task $T_{i+1}$, we start from $\theta$ and do multiple[4] meta-gradient steps with adaptive step sizes as follows (assuming the number of steps is $M$):

$$\phi_i^0 := \theta, \quad \boldsymbol{\tau}_\theta^{1:K} \sim P_{T_i}(\boldsymbol{\tau} \mid \theta),$$
$$\phi_i^m := \phi_i^{m-1} - \alpha_m \nabla_{\phi_i^{m-1}} L_{T_i}\left(\boldsymbol{\tau}_{i,\phi_i^m}^{1:K}\right), \quad m = 1, \ldots, M-1, \tag{7}$$
$$\phi_{i+1} := \phi_i^{M-1} - \alpha_M \nabla_{\phi_i^{M-1}} L_{T_i}\left(\boldsymbol{\tau}_{i,\phi_i^{M-1}}^{1:K}\right)$$

where $\{\alpha_m\}_{m=1}^M$ is a set of meta-gradient step sizes that are optimized jointly with $\theta$. The computation graph for the meta-update is given in Fig. 1c. The expression for the policy gradient is the same as in (4) but with the expectation is now taken w.r.t. to both $T_i$ and $T_{i+1}$:

$$\nabla_{\theta, \alpha} \mathcal{L}_{T_i, T_{i+1}}(\theta, \alpha) =$$

$$\mathbb{E}_{\substack{\boldsymbol{\tau}_{i,\theta}^{1:K} \sim P_{T_i}(\boldsymbol{\tau}|\theta) \\ \boldsymbol{\tau}_{i+1,\phi} \sim P_{T_{i+1}}(\boldsymbol{\tau}|\phi)}} \left[ L_{T_{i+1}}(\boldsymbol{\tau}_{i+1,\phi}) \left[ \nabla_{\theta, \alpha} \log \pi_\phi(\boldsymbol{\tau}_{i+1,\phi}) + \nabla_\theta \sum_{k=1}^K \log \pi_\theta(\boldsymbol{\tau}_{i,\theta}^k) \right] \right] \tag{8}$$

More details and the analog of the policy gradient theorem for our setting are given in Appendix A.

Note that computing adaptation updates requires interacting with the environment under $\pi_\theta$ while computing the meta-loss, $\mathcal{L}_{T_i, T_{i+1}}$, requires using $\pi_\phi$, and hence, interacting with each task in the sequence twice. This is often impossible at execution time, and hence we use slightly different algorithms at training and execution times.

**Meta-learning at training time.** Once we have access to a distribution over pairs of consecutive tasks[5], $\mathcal{P}(T_{i-1}, T_i)$, we can meta-learn the adaptation updates by optimizing $\theta$ and $\alpha$ jointly with a gradient method, as given in Algorithm 1. We use $\pi_\theta$ to collect trajectories from $T_i$ and $\pi_\phi$ when interacting with $T_{i+1}$. Intuitively, the algorithm is searching for $\theta$ and $\alpha$ such that the adaptation update (7) computed on the trajectories from $T_i$ brings us to a policy, $\pi_\phi$, that is good for solving $T_{i+1}$. The main assumption here is that the trajectories from $T_i$ contain some information about $T_{i+1}$. Note that we treat adaptation steps as part of the computation graph (Fig. 1c) and optimize $\theta$ and $\alpha$ via backpropagation through the entire graph, which requires computing second order derivatives.

**Adaptation at execution time.** Note that to compute unbiased adaptation gradients at training time, we have to collect experience in $T_i$ using $\pi_\theta$. At test time, due to environment nonstationarity, we usually do not have the luxury to access to the same task multiple times. Thus, we keep acting according to $\pi_\phi$ and re-use past experience to compute updates of $\phi$ for each new incoming task (see Algorithm 2). To adjust for the fact that the past experience was collected under a policy different from $\pi_\theta$, we use importance weight correction. In case of single step meta-update, we have:

$$\phi_i := \theta - \alpha \frac{1}{K} \sum_{k=1}^K \left( \frac{\pi_\theta(\boldsymbol{\tau}^k)}{\pi_{\phi_{i-1}}(\boldsymbol{\tau}^k)} \right) \nabla_\theta L_{T_{i-1}}(\boldsymbol{\tau}^k), \quad \boldsymbol{\tau}^{1:K} \sim P_{T_{i-1}}(\boldsymbol{\tau} \mid \phi_{i-1}), \tag{9}$$

---

[3]This is due to stability considerations. We find empirically that optimization over sequential updates from $\phi_i$ to $\phi_{i+1}$ is unstable, often tends to diverge, while starting from the same initialization leads to better behavior.
[4]Empirically, it turns out that constructing $\phi$ via multiple meta-gradient steps (between 2 and 5) with adaptive step sizes tends yield better results in practice.
[5]Given a sequences of tasks generated by a nonstationary environment, $T_1, T_2, T_3, \ldots, T_L$, we use the set of all pairs of consecutive tasks, $\{(T_{i-1}, T_i)\}_{i=1}^L$, as the training distribution.

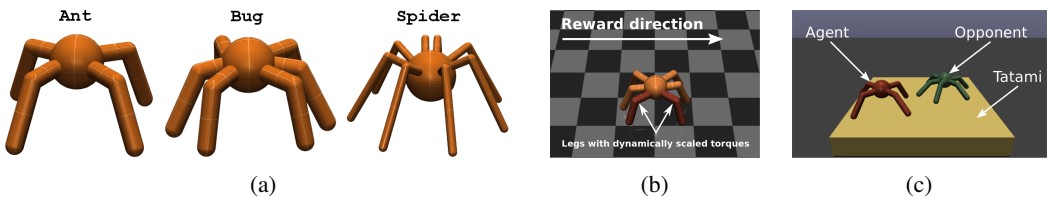

Fig. 2: (a) The three types of agents used in experiments. The robots differ in the anatomy: the number of legs, their positions, and constraints on the thigh and knee joints. (b) The nonstationary locomotion environment. The torques applied to red-colored legs are scaled by a dynamically changing factor. (c) `RoboSumo` environment.

where $\pi_{\phi_{i-1}}$ and $\pi_{\phi_i}$ are used to rollout from $T_{i-1}$ and $T_i$, respectively. Extending importance weight correction to multi-step updates is straightforward and requires simply adding importance weights to each of the intermediate steps in (7).

## 4 ENVIRONMENTS

We have designed a set of environments for testing different aspects of continuous adaptation methods in two scenarios: (i) simple environments that change from episode to episode according to some underlying dynamics, and (ii) a competitive multi-agent environment, `RoboSumo`, that allows different agents to play sequences of games against each other and keep adapting to incremental changes in each other's policies. All our environments are based on MuJoCo physics simulator (Todorov et al., 2012), and all agents are simple multi-leg robots, as shown in Fig. 2a.

### 4.1 DYNAMIC

First, we consider the problem of robotic locomotion in a changing environment. We use a six-leg agent (Fig. 2b) that observes the absolute position and velocity of its body, the angles and velocities of its legs, and it acts by applying torques to its joints. The agent is rewarded proportionally to its moving speed in a fixed direction. To induce nonstationarity, we select a pair of legs of the agent and scale down the torques applied to the corresponding joints by a factor that linearly changes from 1 to 0 over the course of 7 episodes. In other words, during the first episode all legs are fully functional, while during the last episode the agent has two legs fully paralyzed (even though the policy can generate torques, they are multiplied by 0 before being passed to the environment). The goal of the agent is to learn to adapt from episode to episode by changing its gait so that it is able to move with a maximal speed in a given direction despite the changes in the environment (cf. Cully et al., 2015). Also, there are 15 ways to select a pair of legs of a six-leg creature which gives us 15 different nonstationary environments. This allows us to use a subset of these environments for training and a separate held out set for testing. The training and testing procedures are described in the next section.

### 4.2 COMPETITIVE

Our multi-agent environment, `RoboSumo`, allows agents to compete in the 1-vs-1 regime following the standard sumo rules[6]. We introduce three types of agents, `Ant`, `Bug`, and `Spider`, with different anatomies (Fig. 2a). During the game, each agent observes positions of itself and the opponent, its own joint angles, the corresponding velocities, and the forces exerted on its own body (i.e., equivalent of tactile senses). The action spaces are continuous.

**Iterated adaptation games.** To test adaptation, we define the *iterated adaptation game* (Fig. 3)—a game between a pair of agents that consists of $K$ rounds each of which consists of one or more fixed length episodes (500 time steps each). The outcome of each round is either win, loss, or draw. The agent that wins the majority of rounds (with at least 5% margin) is declared the winner of the game. There are two distinguishing aspects of our setup: First, the agents are trained either via pure self-play or versus opponents from a fixed training collection. At test time, they face a new opponent from a testing collection. Second, the agents are allowed to learn (or adapt) at test time. In particular, an

---

[6]To win, the agent has to push the opponent out of the ring or make the opponent's body touch the ground.

Fig. 3: An agent competes with an opponent in an iterated adaptation games that consist of multi-episode rounds. The agent wins a round if it wins the majority of episodes (wins and losses illustrated with color). Both the agent and its opponent may update their policies from round to round (denoted by the version number).

agent should exploit the fact that it plays against the same opponent multiple consecutive rounds and try to adjust its behavior accordingly. Since the opponent may also be adapting, the setup allows to test different continuous adaptation strategies, one versus the other.

**Reward shaping.** In `RoboSumo`, rewards are naturally sparse: the winner gets +2000, the loser is penalized for -2000, and in case of a draw both opponents receive -1000 points. To encourage fast learning at the early stages of training, we shape the rewards given to agents in the following way: the agent (i) gets reward for staying closer to the center of the ring, for moving towards the opponent, and for exerting forces on the opponent's body, and (ii) gets penalty inversely proportional to the opponent's distance to the center of the ring. At test time, the agents continue having access to the shaped reward as well and may use it to update their policies. Throughout our experiments, we use discounted rewards with the discount factor, $\gamma = 0.995$. More details are in Appendix D.2.

**Calibration.** To study adaptation, we need a well-calibrated environment in which none of the agents has an initial advantage. To ensure the balance, we increased the mass of the weaker agents (`Ant` and `Spider`) such that the win rates in games between one agent type versus the other type in the non-adaptation regime became almost equal (for details on calibration see Appendix D.3).

## 5 EXPERIMENTS

Our goal is to test different adaptation strategies in the proposed nonstationary RL settings. However, it is known that the test-time behavior of an agent may highly depend on a variety of factors besides the chosen adaptation method, including training curriculum, training algorithm, policy class, etc. Hence, we first describe the precise setup that we use in our experiments to eliminate irrelevant factors and focus on the effects of adaptation. Most of the low-level details are deferred to appendices. Video highlights of our experiments are available at `https://goo.gl/tboqaN`.

### 5.1 THE SETUP

**Policies.** We consider 3 types of policy networks: (i) a 2-layer MLP, (ii) embedding (i.e., 1 fully-connected layer replicated across the time dimension) followed by a 1-layer LSTM, and (iii) RL[2] (Duan et al., 2016) of the same architecture as (ii) which additionally takes previous reward and done signals as inputs at each step, keeps the recurrent state throughout the entire interaction with a given environment (or an opponent), and resets the state once the latter changes. For advantage functions, we use networks of the same structure as for the corresponding policies and have no parameter sharing between the two. Our meta-learning agents use the same policy and advantage function structures as the baselines and learn a 3-step meta-update with adaptive step sizes as given in (7). Illustrations and details on the architectures are given in Appendix B.

**Meta-learning.** We compute meta-updates via gradients of the negative discounted rewards received during a number of previous interactions with the environment. At training time, meta-learners interact with the environment twice, first using the initial policy, $\pi_\theta$, and then the meta-updated policy, $\pi_\phi$. At test time, the agents are limited to interacting with the environment only once, and hence always act according to $\pi_\phi$ and compute meta-updates using importance-weight correction (see Sec. 3.2 and Algorithm 2). Additionally, to reduce the variance of the meta-updates at test time, the agents store the experience collected during the interaction with the test environment (and the corresponding importance weights) into the experience buffer and keep re-using that experience

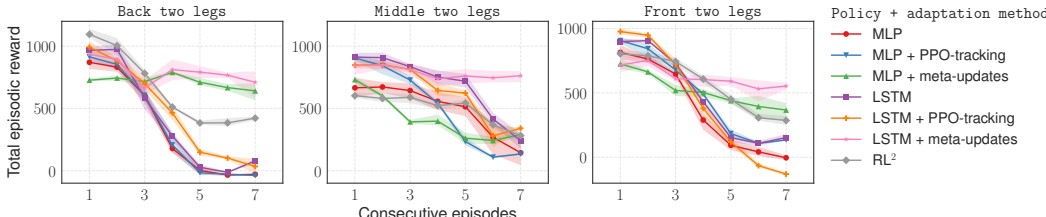

Fig. 4: Episodic rewards for 7 consecutive episodes in 3 held out nonstationary locomotion environments. To evaluate adaptation strategies, we ran each of them in each environment for 7 episodes followed by a full reset of the environment, policy, and meta-updates (repeated 50 times). Shaded regions are 95% confidence intervals. Best viewed in color.

to update $\pi_\phi$ as in (7). The size of the experience buffer is fixed to 3 episodes for nonstationary locomotion and 75 episodes for `RoboSumo`. More details are given in Appendix C.1.

**Adaptation baselines.** We consider the following three baseline strategies:

   (i) naive (or no adaptation),

  (ii) implicit adaptation via $RL^2$, and

 (iii) adaptation via tracking (Sutton et al., 2007) that keeps doing PPO updates at execution time.

**Training in nonstationary locomotion.** We train all methods on the same collection of nonstationary locomotion environments constructed by choosing all possible pairs of legs whose joint torques are scaled except 3 pairs that are held out for testing (i.e., 12 training and 3 testing environments for the six-leg creature). The agents are trained on the environments concurrently, i.e., to compute a policy update, we rollout from all environments in parallel and then compute, aggregate, and average the gradients (for details, see Appendix C.2). LSTM policies retain their state over the course of 7 episodes in each environment. Meta-learning agents compute meta-updates for each nonstationary environment separately.

**Training in `RoboSumo`.** To ensure consistency of the training curriculum for all agents, we first pre-train a number of policies of each type for every agent type via pure self-play with the PPO algorithm (Schulman et al., 2017; Bansal et al., 2018). We snapshot and save versions of the pre-trained policies at each iteration. This lets us train other agents to play against versions of the pre-trained opponents at various stages of mastery. Next, we train the baselines and the meta-learning agents against the pool of pre-trained opponents[7] concurrently. At each iteration $k$ we (a) randomly select an opponent from the training pool, (b) sample a version of the opponent's policy to be in $[1, k]$ (this ensures that even when the opponent is strong, sometimes an undertrained version is selected which allows the agent learn to win at early stages), and (c) rollout against that opponent. All baseline policies are trained with PPO; meta-learners also used PPO as the outer loop for optimizing $\theta$ and $\alpha$ parameters. We retain the states of the LSTM policies over the course of interaction with the same version of the same opponent and reset it each time the opponent version is updated. Similarly to the locomotion setup, meta-learners compute meta-updates for each opponent in the training pool separately. A more detailed description of the distributed training is given in Appendix C.2.

**Experimental design.** We design our experiments to answer the following questions:

- When the interaction with the environment before it changes is strictly limited to one or very few episodes, what is the behavior of different adaptation methods in nonstationary locomotion and competitive multi-agent environments?

- What is the sample complexity of different methods, i.e., how many episodes is required for a method to successfully adapt to the changes? We test this by controlling the amount of experience the agent is allowed to get form the same environment before it changes.

---

[7]In competitive multi-agent environments, besides self-play, there are plenty of ways to train agents, e.g., train them in pairs against each other concurrently, or randomly match and switch opponents each few iterations. We found that concurrent training often leads to an unbalanced population of agents that have been trained under vastly different curricula and introduces spurious effects that interfere with our analysis of adaptation. Hence, we leave the study of adaptation in naturally emerging curricula in multi-agent settings to the future work.

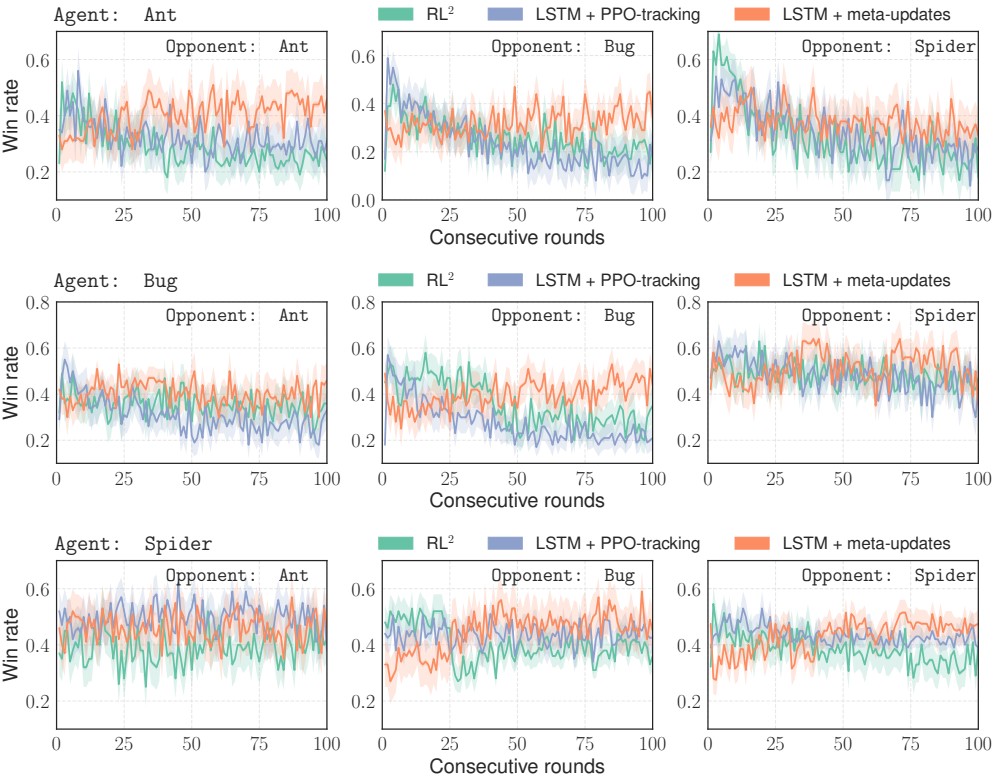

Fig. 5: Win rates for different adaptation strategies in iterated games versus 3 different pre-trained opponents. At test time, both agents and opponents started from versions 700. Opponents' versions were increasing with each consecutive round as if they were learning via self-play, while agents were allowed to adapt only from the limited experience with a given opponent. Each round consisted of 3 episodes. Each iterated game was repeated 100 times; shaded regions denote bootstrapped 95% confidence intervals; no smoothing. Best viewed in color.

Additionally, we ask the following questions specific to the competitive multi-agent setting:

- Given a diverse population of agents that have been trained under the same curriculum, how do different adaptation methods rank in a competition versus each other?

- When the population of agents is evolved for several generations—such that the agents interact with each other via iterated adaptation games, and those that lose disappear while the winners get duplicated—what happens with the proportions of different agents in the population?

## 5.2 ADAPTATION IN THE FEW-SHOT REGIME AND SAMPLE COMPLEXITY

**Few-shot adaptation in nonstationary locomotion environments.** Having trained baselines and meta-learning policies as described in Sec. 5.1, we selected 3 testing environments that corresponded to disabling 3 different pairs of legs of the six-leg agent: back, middle, and front legs. The results are presented on Fig. 4. Three observations: First, during the very first episode, the meta-learned initial policy, $\pi_{\theta^\star}$, turns out to be suboptimal for the task (it underperforms compared to other policies). However, after 1-2 episodes (and environment changes), it starts performing on par with other policies. Second, by the 6th and 7th episodes, meta-updated policies perform much better than the rest. Note that we use 3 gradient meta-updates for the adaptation of the meta-learners; the meta-updates are computed based on experience collected during the previous 2 episodes. Finally, tracking is not able to improve upon the baseline without adaptation and sometimes leads to even worse results.

**Adaptation in RoboSumo under the few-shot constraint.** To evaluate different adaptation methods in the competitive multi-agent setting consistently, we consider a variation of the iterated adaptation game, where changes in the opponent's policies at test time are pre-determined but unknown to the

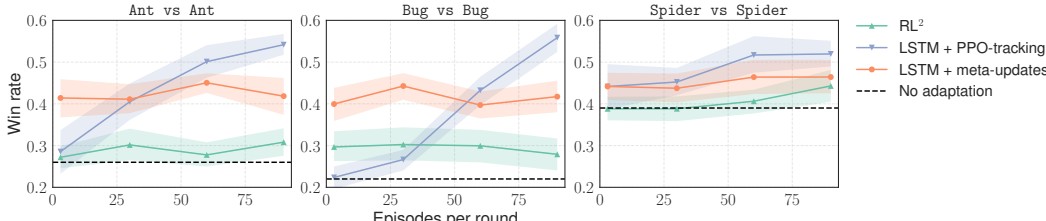

Fig. 6: The effect of increased number of episodes per round in the iterated games versus a learning opponent.

agents. In particular, we pre-train 3 opponents (1 of each type, Fig. 2a) with LSTM policies with PPO via self-play (the same way as we pre-train the training pool of opponents, see Sec. 5.1) and snapshot their policies at each iteration. Next, we run iterated games between our trained agents that use different adaptation algorithms versus policy snapshots of the pre-trained opponents. Crucially, the policy version of the opponent keeps increasing from round to round as if it was training via self-play[8]. The agents have to keep adapting to increasingly more competent versions of the opponent (see Fig. 3). This setup allows us to test different adaptation strategies consistently against the same learning opponents.

The results are given on Fig. 5. We note that meta-learned adaptation strategies, in most cases, are able to adapt and improve their win-rates within about 100 episodes of interaction with constantly improving opponents. On the other hand, performance of the baselines often deteriorates during the rounds of iterated games. Note that the pre-trained opponents were observing 90 episodes of self-play per iteration, while the agents have access to only 3 episodes per round.

**Sample complexity of adaptation in `RoboSumo`.** Meta-learning helps to find an update suitable for fast or few-shot adaptation. However, how do different adaptation methods behave when more experience is available? To answer this question, we employ the same setup as previously and vary the number of episodes per round in the iterated game from 3 to 90. Each iterated game is repeated 20 times, and we measure the win-rates during the last 25 rounds of the game.

The results are presented on Fig. 6. When the number of episodes per round goes above 50, adaptation via tracking technically turns into "learning at test time," and it is able to learn to compete against the self-trained opponents that it has never seen at training time. The meta-learned adaptation strategy performed near constantly the same in both few-shot and standard regimes. This suggests that the meta-learned strategy acquires a particular bias at training time that allows it to perform better from limited experience but also limits its capacity of utilizing more data. Note that, by design, the meta-updates are fixed to only 3 gradient steps from $\theta^\star$ with step-sizes $\alpha^\star$ (learned at training), while tracking keeps updating the policy with PPO throughout the iterated game. Allowing for meta-updates that become more flexible with the availability of data can help to overcome this limitation. We leave this to future work.

## 5.3 Evaluation on the population-level

Combining different adaptation strategies with different policies and agents of different morphologies puts us in a situation where we have a diverse population of agents which we would like to rank according to the level of their mastery in adaptation (or find the "fittest"). To do so, we employ TrueSkill (Herbrich et al., 2007)—a metric similar to the ELO rating, but more popular in 1-vs-1 competitive video-games.

In this experiment, we consider a population of 105 trained agents: 3 agent types, 7 different policy and adaptation combinations, and 5 different stages of training (from 500 to 2000 training iterations). First, we assume that the initial distribution of any agent's skill is $\mathcal{N}(25, 25/3)$ and the default distance that guarantees about 76% of winning, $\beta = 4.1667$. Next, we randomly generate 1000 matches between pairs of opponents and let them adapt while competing with each other in 100-round iterated adaptation games (states of the agents are reset before each game). After each game, we

---

[8]At the beginning of the iterated game, both agents and their opponent start from version 700, i.e., from the policy obtained after 700 iterations (PPO epochs) of learning to ensure that the initial policy is reasonable.

record the outcome and updated our belief about the skill of the corresponding agents using the TrueSkill algorithm[9]. The distributions of the skill for the agents of each type after 1000 iterated adaptation games between randomly selected players from the pool are visualized in Fig. 7.

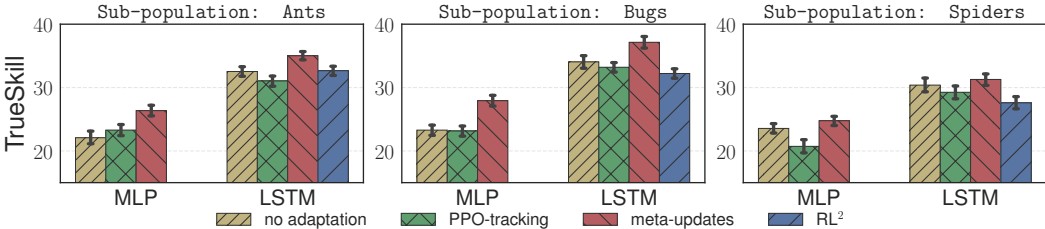

Fig. 7: TrueSkill for the top-performing MLP- and LSTM-based agents. TrueSkill was computed based on outcomes (win, loss, or draw) in 1000 iterated adaptation games (100 consecutive rounds per game, 3 episodes per round) between randomly selected pairs of opponents from a population of 105 pre-trained agents.

There are a few observations we can make: First, recurrent policies were dominant. Second, adaptation via RL$^2$ tended to perform equally or a little worse than plain LSTM with or without tracking in this setup. Finally, agents that meta-learned adaptation rules at training time, consistently demonstrated higher skill scores in each of the categories corresponding to different policies and agent types.

Finally, we enlarge the population from 105 to 1050 agents by duplicating each of them 10 times and evolve it (in the "natural selection" sense) for several generations as follows. Initially, we start with a balanced population of different creatures. Next, we randomly match 1000 pairs of agents, make them play iterated adaptation games, remove the agents that lost from the population and duplicate the winners. The same process is repeated 10 times. The result is presented in Fig 8. We see that many agents quickly disappear form initially uniform population and the meta-learners end up dominating.

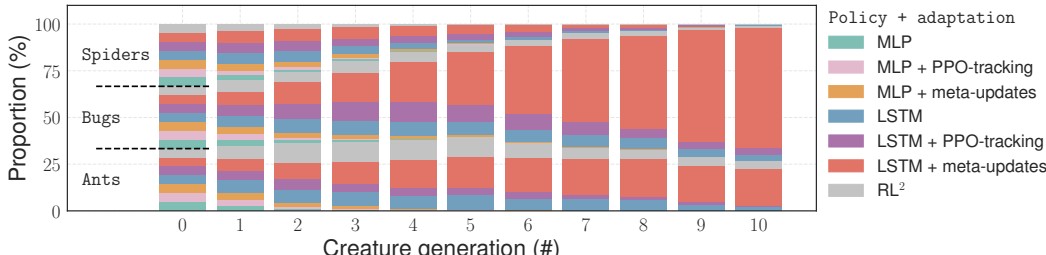

Fig. 8: Evolution of a population of 1050 agents for 10 generations. Best viewed in color.

## 6 CONCLUSION AND FUTURE DIRECTIONS

In this work, we proposed a simple gradient-based meta-learning approach suitable for continuous adaptation in nonstationary environments. The key idea of the method is to regard nonstationarity as a sequence of stationary tasks and train agents to exploit the dependencies between consecutive tasks such that they can handle similar nonstationarities at execution time. We applied our method to nonstationary locomotion and within a competitive multi-agent setting. For the latter, we designed the `RoboSumo` environment and defined iterated adaptation games that allowed us to test various aspects of adaptation strategies. In both cases, meta-learned adaptation rules were more efficient than the baselines in the few-shot regime. Additionally, agents that meta-learned to adapt demonstrated the highest level of skill when competing in iterated games against each other.

The problem of continuous adaptation in nonstationary and competitive environments is far from being solved, and this work is the first attempt to use meta-learning in such setup. Indeed, our meta-learning algorithm has a few limiting assumptions and design choices that we have made mainly due to computational considerations. First, our meta-learning rule is to one-step-ahead update of the

---

[9]We used an implementation from `http://trueskill.org/`.

policy and is computationally similar to backpropagation through time with a unit time lag. This could potentially be extended to fully recurrent meta-updates that take into account the full history of interaction with the changing environment. Additionally, our meta-updates were based on the gradients of a surrogate loss function. While such updates explicitly optimized the loss, they required computing second order derivatives at training time, slowing down the training process by an order of magnitude compared to baselines. Utilizing information provided by the loss but avoiding explicit backpropagation through the gradients would be more appealing and scalable. Finally, our approach is unlikely to work with sparse rewards as the meta-updates use policy gradients and heavily rely on the reward signal. Introducing auxiliary dense rewards designed to enable meta-learning is a potential way to overcome this issue that we would like to explore in the future work.

## ACKNOWLEDGEMENTS

We would like to thank Harri Edwards, Jakob Foerster, Aditya Grover, Aravind Rajeswaran, Vikash Kumar, Yuhuai Wu and many others at OpenAI for helpful comments and fruitful discussions.

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

## A    DERIVATIONS AND THE POLICY GRADIENT THEOREM

In this section, we derive the policy gradient update for MAML as give in (4) as well as formulate and equivalent of the policy gradient theorem (Sutton et al., 2000) in the learning-to-learn setting.

Our derivation is not bound to a particular form of the adaptation update. In general, we are interested in meta-learning a *procedure*, $f_\theta$, parametrized by $\theta$, which, given access to a limited experience on a task, can produce a good policy for solving it. Note that $f_\theta$ is responsible for both collecting the initial experience and constructing the final policy for the given task. For example, in case of MAML (Finn et al., 2017b), $f_\theta$ is represented by the initial policy, $\pi_\theta$, and the adaptation update rule (4) that produces $\pi_\phi$ with $\phi := \theta - \alpha \nabla_\theta L_T(\boldsymbol{\tau}_\theta^{1:K})$.

More formally, after querying $K$ trajectories, $\boldsymbol{\tau}_\theta^{1:K}$, we want to produce $\pi_\phi$ that minimizes the expected loss w.r.t. the distribution over tasks:

$$\mathcal{L}(\theta) := \mathbb{E}_{T \sim \mathcal{D}(T)} \left[ \mathbb{E}_{\boldsymbol{\tau}_\theta^{1:K} \sim p_T(\boldsymbol{\tau}|\theta)} \left[ \mathbb{E}_{\boldsymbol{\tau}_\phi \sim p_T(\boldsymbol{\tau}|\phi)} \left[ L_T(\boldsymbol{\tau}_\phi) \mid \boldsymbol{\tau}_\theta^{1:K} \right] \right] \right] \tag{10}$$

Note that the inner-most expectation is conditional on the experience, $\boldsymbol{\tau}_\theta^{1:K}$, which our meta-learning procedure, $f_\theta$, collects to produce a task-specific policy, $\pi_\phi$. Assuming that the loss $L_T(\boldsymbol{\tau}_\theta^{1:K})$ is linear in trajectories, and using linearity of expectations, we can drop the superscript $1:K$ and denote the trajectory sampled under $\phi_\theta$ for task $T_i$ simply as $\boldsymbol{\tau}_{\theta,i}$. At training time, we are given a finite sample of tasks from the distribution $\mathcal{D}(T)$ and can search for $\hat\theta$ close to optimal by optimizing over the empirical distribution:

$$\hat\theta := \operatorname*{argmin}_\theta \hat{\mathcal{L}}(\theta), \text{ where } \hat{\mathcal{L}}(\theta) := \frac{1}{N} \sum_{i=1}^N \mathbb{E}_{\boldsymbol{\tau}_{\theta,i} \sim p_{T_i}(\boldsymbol{\tau}|\theta)} \left[ \mathbb{E}_{\boldsymbol{\tau}_{\phi,i} \sim p_{T_i}(\boldsymbol{\tau}|\phi)} \left[ L_{T_i}(\boldsymbol{\tau}_{\phi,i}) \mid \boldsymbol{\tau}_{\theta,i} \right] \right] \tag{11}$$

We re-write the objective function for task $T_i$ in (11) more explicitly by expanding the expectations:

$$\mathcal{L}_{T_i}(\theta) := \mathbb{E}_{\boldsymbol{\tau}_{\theta,i} \sim p_{T_i}(\boldsymbol{\tau}|\theta)} \left[ \mathbb{E}_{\boldsymbol{\tau}_{\phi,i} \sim p_{T_i}(\boldsymbol{\tau}|\phi)} \left[ L_{T_i}(\boldsymbol{\tau}_{\phi,i}) \mid \boldsymbol{\tau}_{\theta,i} \right] \right] = $$
$$\int L_{T_i}(\boldsymbol{\tau}_{\phi,i}) \, P_{T_i}(\boldsymbol{\tau}_{\phi,i} \mid \phi) \, P_{T_i}(\phi \mid \theta, \boldsymbol{\tau}_{\theta,i}) \, P_{T_i}(\boldsymbol{\tau}_{\theta,i} \mid \theta) \, d\boldsymbol{\tau}_{\phi,i} \, d\phi \, d\boldsymbol{\tau}_{\theta,i} \tag{12}$$

Trajectories, $\boldsymbol{\tau}_{\phi,i}$ and $\boldsymbol{\tau}_{\theta,i}$, and parameters $\phi$ of the policy $\pi_\phi$ can be thought as random variables that we marginalize out to construct the objective that depends on $\theta$ only. The adaptation update rule (4) assumes the following $P_{T_i}(\phi \mid \theta, \boldsymbol{\tau}_{\theta,i})$:

$$P_{T_i}(\phi \mid \theta, \boldsymbol{\tau}_{\theta,i}) := \delta \left( \theta - \alpha \nabla_\theta \frac{1}{K} \sum_{k=1}^K L_{T_i}(\boldsymbol{\tau}_{\theta,i}^k) \right) \tag{13}$$

Note that by specifying $P_{T_i}(\phi \mid \theta, \boldsymbol{\tau}_{\theta,i})$ differently, we may arrive at different meta-learning algorithms. After plugging (13) into (12) and integrating out $\phi$, we get the following expected loss for task $T_i$ as a function of $\theta$:

$$\mathcal{L}_{T_i}(\theta) = \mathbb{E}_{\boldsymbol{\tau}_{\theta,i} \sim p_{T_i}(\boldsymbol{\tau}|\theta)} \left[ \mathbb{E}_{\boldsymbol{\tau}_{\phi,i} \sim p_{T_i}(\boldsymbol{\tau}|\phi)} \left[ L_{T_i}(\boldsymbol{\tau}_{\phi,i}) \mid \boldsymbol{\tau}_{\theta,i} \right] \right] = $$
$$\int L_{T_i}(\boldsymbol{\tau}_{\phi,i}) \, P_{T_i} \left( \boldsymbol{\tau}_{\phi,i} \mid \theta - \alpha \nabla_\theta \frac{1}{K} \sum_{k=1}^K L_{T_i}(\boldsymbol{\tau}_{\theta,i}^k) \right) \, P_{T_i}(\boldsymbol{\tau}_{\theta,i} \mid \theta) \, d\boldsymbol{\tau}_{\phi,i} \, d\boldsymbol{\tau}_{\theta,i} \tag{14}$$

The gradient of (14) will take the following form:

$$\nabla_\theta \mathcal{L}_{T_i}(\theta) = \int \left[ L_{T_i}(\boldsymbol{\tau}_{\phi,i}) \nabla_\theta \log P_{T_i}(\boldsymbol{\tau}_{\phi,i} \mid \phi) \right] P_{T_i}(\boldsymbol{\tau}_{\phi,i} \mid \phi) \, P_{T_i}(\boldsymbol{\tau}_{\theta,i} \mid \theta) \, d\boldsymbol{\tau} \, d\boldsymbol{\tau}_{\theta,i} + $$
$$\int \left[ L_{T_i}(\boldsymbol{\tau}) \nabla_\theta \log P_{T_i}(\boldsymbol{\tau}_{\theta,i} \mid \theta) \right] P_{T_i}(\boldsymbol{\tau} \mid \phi) \, P_{T_i}(\boldsymbol{\tau}_{\theta,i} \mid \theta) \, d\boldsymbol{\tau} \, d\boldsymbol{\tau}_{\theta,i} \tag{15}$$

where $\phi = \phi(\theta, \boldsymbol{\tau}_{\theta,i}^{1:K})$ as given in (14). Note that the expression consists of two terms: the first term is the standard policy gradient w.r.t. the updated policy, $\pi_\phi$, while the second one is the policy gradient w.r.t. the original policy, $\pi_\phi$, that is used to collect $\boldsymbol{\tau}_{\theta,i}^{1:K}$. If we were to omit marginalization

of $\boldsymbol{\tau}_{\theta,i}^{1:K}$ (as it was done in the original paper (Finn et al., 2017b)), the terms would disappear. Finally, the gradient can be re-written in a more succinct form:

$$\nabla_\theta \mathcal{L}_{T_i}(\theta) = \mathbb{E}_{\substack{\boldsymbol{\tau}_{\theta,i}^{1:K} \sim P_{T_i}(\boldsymbol{\tau}|\theta) \\ \boldsymbol{\tau} \sim P_{T_i}(\boldsymbol{\tau}|\phi)}} \left[ L_{T_i}(\boldsymbol{\tau}) \left[ \nabla_\theta \log \pi_\phi(\boldsymbol{\tau}) + \nabla_\theta \sum_{k=1}^K \log \pi_\theta(\boldsymbol{\tau}_k) \right] \right] \tag{16}$$

The update given in (16) is an unbiased estimate of the gradient as long as the loss $L_{T_i}$ is simply the sum of discounted rewards (i.e., it extends the classical REINFORCE algorithm (Williams, 1992) to meta-learning). Similarly, we can define $L_{T_i}$ that uses a value or advantage function and extend the policy gradient theorem Sutton et al. (2000) to make it suitable for meta-learning.

**Theorem 1** (Meta policy gradient theorem). *For any MDP, gradient of the value function w.r.t. $\theta$ takes the following form:*

$$\nabla_\theta V_T^\theta(\mathbf{x}_0) =$$

$$\mathbb{E}_{\boldsymbol{\tau}_{1:K} \sim p_T(\boldsymbol{\tau}|\theta)} \left[ \sum_{\mathbf{x}} d_T^\phi(\mathbf{x}) \sum_a \frac{\partial \pi_\phi(a \mid \mathbf{x})}{\partial \theta} Q_T^\phi(a, \mathbf{x}) \right] +$$

$$\mathbb{E}_{\boldsymbol{\tau}_{1:K} \sim p_T(\boldsymbol{\tau}|\theta)} \left[ \left( \frac{\partial}{\partial \theta} \sum_{k=1}^K \log \pi_\theta(\boldsymbol{\tau}_k) \right) \sum_a \pi_\phi(a \mid \mathbf{x}_0) Q_T^\phi(a, \mathbf{x}_0) \right], \tag{17}$$

*where $d_T^\phi(\mathbf{x})$ is the stationary distribution under policy $\pi_\phi$.*

**Proof.** We define task-specific value functions under the generated policy, $\pi_\phi$, as follows:

$$V_T^\phi(\mathbf{x}_0) = \mathbb{E}_{\boldsymbol{\tau} \sim p_T(\boldsymbol{\tau}|\phi)} \left[ \sum_{t=k}^H \gamma^t R_T(\mathbf{x}_t) \mid \mathbf{x}_0 \right],$$

$$Q_T^\phi(\mathbf{x}_0, a_0) = \mathbb{E}_{\boldsymbol{\tau} \sim p_T(\boldsymbol{\tau}|\phi)} \left[ \sum_{t=k}^H \gamma^t R_T(\mathbf{x}_t) \mid \mathbf{x}_0, a_0 \right], \tag{18}$$

where the expectations are taken w.r.t. the dynamics of the environment of the given task, $T$, and the policy, $\pi_\phi$. Next, we need to marginalize out $\boldsymbol{\tau}_{1:K}$:

$$V_T^\theta(\mathbf{x}_0) = \mathbb{E}_{\boldsymbol{\tau}_{1:K} \sim p_T(\boldsymbol{\tau}|\theta)} \left[ \mathbb{E}_{\boldsymbol{\tau} \sim p_T(\boldsymbol{\tau}|\phi)} \left[ \sum_{t=k}^H \gamma^t R_T(\mathbf{x}_t) \mid \mathbf{x}_0 \right] \right], \tag{19}$$

and after the gradient w.r.t. $\theta$, we arrive at:

$$\nabla_\theta V_T^\theta(\mathbf{x}_0) =$$

$$\mathbb{E}_{\boldsymbol{\tau}_{1:K} \sim p_T(\boldsymbol{\tau}|\theta)} \left[ \sum_a \frac{\partial \pi_\phi(a \mid \mathbf{x}_0)}{\partial \theta} Q_T^\phi(a, \mathbf{x}_0) + \pi_\phi(a \mid \mathbf{x}_0) \frac{\partial Q_T^\phi(a, \mathbf{x}_0)}{\partial \theta} \right] +$$

$$\mathbb{E}_{\boldsymbol{\tau}_{1:K} \sim p_T(\boldsymbol{\tau}|\theta)} \left[ \left( \sum_{k=1}^K \frac{\partial}{\partial \theta} \log \pi_\theta(\boldsymbol{\tau}_k) \right) \sum_a \pi_\phi(a \mid \mathbf{x}_0) Q_T^\phi(a, \mathbf{x}_0) \right], \tag{20}$$

where the first term is similar to the expression used in the original policy gradient theorem (Sutton et al., 2000) while the second one comes from differentiating trajectories $\boldsymbol{\tau}_{1:K}$ that depend on $\theta$. Following Sutton et al. (2000), we unroll the derivative of the Q-function in the first term and arrive at the following final expression for the policy gradient:

$$\nabla_\theta V_T^\theta(\mathbf{x}_0) =$$

$$\mathbb{E}_{\boldsymbol{\tau}_{1:K} \sim p_T(\boldsymbol{\tau}|\theta)} \left[ \sum_{\mathbf{x}} d_T^\phi(\mathbf{x}) \sum_a \frac{\partial \pi_\phi(a \mid \mathbf{x})}{\partial \theta} Q_T^\phi(a, \mathbf{x}) \right] +$$

$$\mathbb{E}_{\boldsymbol{\tau}_{1:K} \sim p_T(\boldsymbol{\tau}|\theta)} \left[ \left( \frac{\partial}{\partial \theta} \sum_{k=1}^K \log \pi_\theta(\boldsymbol{\tau}_k) \right) \sum_a \pi_\phi(a \mid \mathbf{x}_0) Q_T^\phi(a, \mathbf{x}_0) \right] \tag{21}$$

$\square$

**Remark 1.** *The same theorem is applicable to the continuous setting with the only changes in the distributions used to compute expectations in (17) and (18). In particular, the outer expectation in (17) should be taken w.r.t. $p_{T_i}(\boldsymbol{\tau} \mid \theta)$ while the inner expectation w.r.t. $p_{T_{i+1}}(\boldsymbol{\tau} \mid \phi)$.*

## A.1 MULTIPLE ADAPTATION GRADIENT STEPS

All our derivations so far assumed single step gradient-based adaptation update. Experimentally, we found that the multi-step version of the update often leads to a more stable training and better test time performance. In particular, we construct $\phi$ via intermediate $M$ gradient steps:

$$
\begin{aligned}
\phi^0 &:= \theta, \quad \boldsymbol{\tau}_\theta^{1:K} \sim P_T(\boldsymbol{\tau} \mid \theta), \\
\phi^m &:= \phi^{m-1} - \alpha_m \nabla_{\phi^{m-1}} L_T\left(\boldsymbol{\tau}_{\phi^{m-1}}^{1:K}\right), \quad m = 1, \dots, M-1, \\
\phi &:= \phi^{M-1} - \alpha_M \nabla_{\phi^{M-1}} L_T\left(\boldsymbol{\tau}_{\phi^{M-1}}^{1:K}\right)
\end{aligned}
\tag{22}
$$

where $\phi^m$ are intermediate policy parameters. Note that each intermediate step, $m$, requires interacting with the environment and sampling intermediate trajectories, $\boldsymbol{\tau}_{\phi^m}^{1:K}$. To compute the policy gradient, we need to marginalize out all the intermediate random variables, $\pi_{\phi^m}$ and $\boldsymbol{\tau}_{\phi^m}^{1:K}$, $m = 1, \dots, M$. The objective function (12) takes the following form:

$$
\begin{aligned}
\mathcal{L}_T(\theta) = \\
\int L_T(\boldsymbol{\tau}) \, P_T\left(\boldsymbol{\tau} \mid \phi\right) \, P_T\left(\phi \mid \phi^{M-1}, \boldsymbol{\tau}_{\phi^{M-1}}^{1:K}\right) d\boldsymbol{\tau} d\phi \times \\
\prod_{m=1}^{M-2} P_T\left(\boldsymbol{\tau}_{\phi^{m+1}}^{1:K} \mid \phi^{m+1}\right) P_T\left(\phi^{m+1} \mid \phi^m, \boldsymbol{\tau}_{\phi^m}^{1:K}\right) d\boldsymbol{\tau}_{\phi^{m+1}}^{1:K} d\phi^{m+1} \times \\
P_T\left(\boldsymbol{\tau}^{1:K} \mid \theta\right) d\boldsymbol{\tau}^{1:K}
\end{aligned}
\tag{23}
$$

Since $P_T\left(\phi^{m+1} \mid \phi^m, \boldsymbol{\tau}_{\phi^m}^{1:K}\right)$ at each intermediate steps are delta functions, the final expression for the multi-step MAML objective has the same form as (14), with integration taken w.r.t. all intermediate trajectories. Similarly, an unbiased estimate of the gradient of the objective gets $M$ additional terms:

$$
\nabla_\theta \mathcal{L}_T = \mathbb{E}_{\{\boldsymbol{\tau}_{\phi^m}^{1:K}\}_{m=0}^{M-1}, \boldsymbol{\tau}}\left[ L_T(\boldsymbol{\tau}) \left[ \nabla_\theta \log \pi_\phi(\boldsymbol{\tau}) + \sum_{m=0}^{M-1} \nabla_\theta \sum_{k=1}^{K} \log \pi_{\phi^m}(\boldsymbol{\tau}_{\phi^m}^k) \right] \right],
\tag{24}
$$

where the expectation is taken w.r.t. trajectories (including all intermediate ones). Again, note that at training time we do not constrain the number of interactions with each particular environment and do rollout using each intermediate policy to compute updates. At testing time, we interact with the environment only once and rely on the importance weight correction as described in Sec. 3.2.

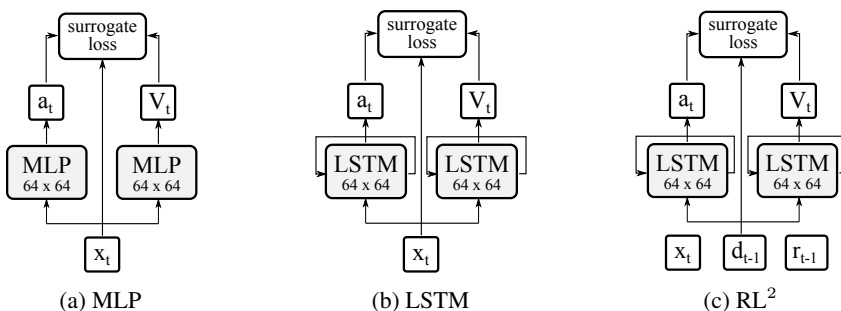

Fig. 9: Policy and value function architectures.

# B    ADDITIONAL DETAILS ON THE ARCHITECTURES

The neural architectures used for our policies and value functions are illustrated in Fig. 9. Our MLP architectures were memory-less and reactive. The LSTM architectures had used a fully connected embedding layer (with 64 hidden units) followed by a recurrent layer (also with 64 units). The state in LSTM-based architectures was kept throughout each episode and reset to zeros at the beginning of each new episode. The RL$^2$ architecture additionally took reward and done signals from the previous time step and kept the state throughout the whole interactions with a given environment (or opponent). The recurrent architectures were unrolled for $T = 10$ time steps and optimized with PPO via backprop through time.

# C    ADDITIONAL DETAILS ON META-LEARNING AND OPTIMIZATION

## C.1    META-UPDATES FOR CONTINUOUS ADAPTATION

Our meta-learned adaptation methods were used with MLP and LSTM policies (Fig. 9). The meta-updates were based on 3 gradient steps with adaptive step sizes $\alpha$ were initialized with 0.001. There are a few additional details to note:

1. $\theta$ and $\phi$ parameters were a concatenation of the policy and the value function parameters.
2. At the initial stages of optimization, meta-gradient steps often tended to "explode", hence we clipped them by values norms to be between -0.1 and 0.1.
3. We used different surrogate loss functions for the meta-updates and for the outer optimization. For meta-updates, we used the vanilla policy gradients computed on the negative discounted rewards, while for the outer optimization loop we used the PPO objective.

## C.2    ON PPO AND ITS DISTRIBUTED IMPLEMENTATION

As mentioned in the main text and similar to (Bansal et al., 2018), large batch sizes were used to ensure enough exploration throughout policy optimization and were critical for learning in the competitive setting of RoboSumo. In our experiments, the epoch size of the PPO was set 32,000 episodes and the batch size was set to 8,000. The PPO clipping hyperparameter was set to $\epsilon = 0.2$ and the KL penalty was set to 0. In all our experiments, the learning rate (for meta-learning, the learning rate for $\theta$ and $\alpha$) was set to 0.0003. The generalized advantage function estimator (GAE) (Schulman et al., 2015b) was optimized jointly with the policy (we used $\gamma = 0.995$ and $\lambda = 0.95$).

To train our agents in reasonable time, we used a distributed implementation of the PPO algorithm. To do so, we versioned the agent's parameters (i.e., kept parameters after each update and assigned it a version number) and used a versioned queue for rollouts. Multiple worker machines were generating rollouts in parallel for the most recent available version of the agent parameters and were pushing them into the versioned rollout queue. The optimizer machine collected rollouts from the queue and made a PPO optimization steps (see (Schulman et al., 2017) for details) as soon as enough rollouts were available.

We trained agents on multiple environments simultaneously. In nonstationary locomotion, each environment corresponded to a different pair of legs of the creature becoming dysfunctional. In `RoboSumo`, each environment corresponded to a different opponent in the training pool. Simultaneous training was achieved via assigning these environments to rollout workers uniformly at random, so that the rollouts in each mini-batch were guaranteed to come from all training environments.

## D    ADDITIONAL DETAILS ON THE ENVIRONMENTS

### D.1    OBSERVATION AND ACTION SPACES

Both observation and action spaces in `RoboSumo` continuous. The observations of each agent consist of the position of its own body (7 dimensions that include 3 absolute coordinates in the global cartesian frame and 4 quaternions), position of the opponent's body (7 dimensions), its own joint angles and velocities (2 angles and 2 velocities per leg), and forces exerted on each part of its own body (6 dimensions for torso and 18 for each leg) and forces exerted on the opponent's torso (6 dimensions). All forces were squared and clipped at 100. Additionally, we normalized observations using a running mean and clipped their values between -5 and 5. The action spaces had 2 dimensions per joint. Table 1 summarizes the observation and action spaces for each agent type.

Table 1: Dimensionality of the observation and action spaces of the agents in `RoboSumo`.

| Agent | Observation space | | | | | Action space |
| | Self | | | Opponent | | |
| | Coordinates | Velocities | Forces | Coordinates | Forces | |
|---|---|---|---|---|---|---|
| Ant | 15 | 14 | 78 | 7 | 6 | 8 |
| Bug | 19 | 18 | 114 | 7 | 6 | 12 |
| Spider | 23 | 22 | 150 | 7 | 6 | 16 |

Note that the agents observe neither any of the opponents velocities, nor positions of the opponent's limbs. This allows us to keep the observation spaces consistent regardless of the type of the opponent. However, even though the agents are blind to the opponent's limbs, they can sense them via the forces applied to the agents' bodies when in contact with the opponent.

### D.2    SHAPED REWARDS

In `RoboSumo`, the winner gets 2000 reward, the loser is penalized for -2000, and in case of draw both agents get -1000. In addition to the sparse win/lose rewards, we used the following dense rewards to encourage fast learning at the early training stages:

- **Quickly push the opponent outside.** The agent got penalty at each time step proportional to $\exp\{-d_{\text{opp}}\}$ where $d_{\text{opp}}$ was the distance of the opponent from the center of the ring.

- **Moving towards the opponent.** Reward at each time step proportional to magnitude of the velocity component towards the opponent.

- **Hit the opponent.** Reward proportional to the square of the total forces exerted on the opponent's torso.

- **Control penalty.** The $l_2$ penalty on the actions to prevent jittery/unnatural movements.

### D.3    ROBOSUMO CALIBRATION

To calibrate the `RoboSumo` environment we used the following procedure. First, we trained each agent via pure self-play with LSTM policy using PPO for the same number of iterations, tested them one against the other (without adaptation), and recorded the win rates (Table 2). To ensure the balance, we kept increasing the mass of the weaker agents and repeated the calibration procedure until the win rates equilibrated.

Table 2: Win rates for the first agent in the 1-vs-1 `RoboSumo` *without adaptation* before and after calibration.

| Masses (`Ant`, `Bug`, `Spider`) | `Ant` vs. `Bug` | `Ant` vs. `Spider` | `Bug` vs. `Spider` |
|---|---|---|---|
| Initial (10, 10, 10) | $25.2 \pm 3.9\%$ | $83.6 \pm 3.1\%$ | $90.2 \pm 2.7\%$ |
| Calibrated (13, 10, 39) | $50.6 \pm 5.6\%$ | $51.6 \pm 3.4\%$ | $51.7 \pm 2.8\%$ |

# E  ADDITIONAL DETAILS ON EXPERIMENTS

## E.1  AVERAGE WIN RATES

Table 3 gives average win rates for the last 25 rounds of iterated adaptation games played by different agents with different adaptation methods (win rates for each episode are visualized in Figure 5).

Table 3: Average win-rates (95% CI) in the last 25 rounds of the 100-round iterated adaptation games between different agents and different opponents. The base policy and value function were LSTMs with 64 hidden units.

| Agent | Opponent | Adaptation Strategy | | |
|---|---|---|---|---|
| | | RL$^2$ | LSTM + PPO-tracking | LSTM + meta-updates |
| Ant | `Ant` | $24.9\ (5.4)\%$ | $30.0\ (6.7)\%$ | $\mathbf{44.0}\ (7.7)\%$ |
| | `Bug` | $21.0\ (6.3)\%$ | $15.6\ (7.1)\%$ | $\mathbf{34.6}\ (8.1)\%$ |
| | `Spider` | $24.8\ (10.5)\%$ | $27.6\ (8.4)\%$ | $35.1\ (7.7)\%$ |
| Bug | `Ant` | $33.5\ (6.9)\%$ | $26.6\ (7.4)\%$ | $\mathbf{39.5}\ (7.1)\%$ |
| | `Bug` | $28.6\ (7.4)\%$ | $21.2\ (4.2)\%$ | $\mathbf{43.7}\ (8.0)\%$ |
| | `Spider` | $45.8\ (8.1)\%$ | $42.6\ (12.9)\%$ | $52.0\ (13.9)\%$ |
| Spider | `Ant` | $40.3\ (9.7)\%$ | $48.0\ (9.8)\%$ | $45.3\ (10.9)\%$ |
| | `Bug` | $38.4\ (7.2)\%$ | $43.9\ (7.1)\%$ | $48.4\ (9.2)\%$ |
| | `Spider` | $33.9\ (7.2)\%$ | $42.2\ (3.9)\%$ | $46.7\ (3.8)\%$ |

## E.2  TRUESKILL RANK OF THE TOP AGENTS

| Rank | Agent | TrueSkill rank* |
|---|---|---|
| 1 | Bug + LSTM-meta | 31.7 |
| 2 | Ant + LSTM-meta | 30.8 |
| 3 | Bug + LSTM-track | 29.1 |
| 4 | Ant + RL$^2$ | 28.6 |
| 5 | Ant + LSTM | 28.4 |
| 6 | Bug + MLP-meta | 23.4 |
| 7 | Ant + MLP-meta | 21.6 |
| 8 | Spider + MLP-meta | 20.5 |
| 9 | Spider + MLP | 19.0 |
| 10 | Bug + MLP-track | 18.9 |

\* The rank is a conservative estimate of the skill, $r = \mu - 3\sigma$, to ensure that the actual skill of the agent is higher with 99% confidence.

Table 4 & Fig. 10: Top-5 agents with MLP and LSTM policies from the population ranked by TrueSkill. The heatmap shows *a priori* win-rates in iterated games based on TrueSkill for the top agents against each other.

Since TrueSkill represents the belief about the skill of an agent as a normal distribution (i.e., with two parameters, $\mu$ and $\sigma$), we can use it to infer *a priori* probability of an agent, $a$, winning against

its opponent, $o$, as follows (Herbrich et al., 2007):

$$P(a \text{ wins } o) = \Phi\left(\frac{\mu_a - \mu_o}{\sqrt{2\beta^2 + \sigma_a^2 + \sigma_o^2}}\right), \text{ where } \Phi(x) := \frac{1}{2}\left[1 + \text{erf}\left(\frac{x}{\sqrt{2}}\right)\right] \qquad (25)$$

The ranking of the top-5 agents with MLP and LSTM policies according to their TrueSkill is given in Tab. 1 and the *a priori* win rates in Fig. 10. Note that within the LSTM and MLP categories, the best meta-learners are 10 to 25% more likely to win the best agents that use other adaptation strategies.

### E.3 INTOLERANCE TO LARGE DISTRIBUTIONAL SHIFTS

Continuous adaptation via meta-learning assumes consistency in the changes of the environment or the opponent. What happens if the changes are drastic? Unfortunately, the training process of our meta-learning procedure turns out to be sensitive to such shifts and can diverge when the distributional shifts from iteration to iteration are large. Fig. 11 shows the training curves for a meta-learning agent with MLP policy trained against versions of an MLP opponent pre-trained via self-play. At each iteration, we kept updating the opponent policy by 1 to 10 steps. The meta-learned policy was able to achieve non-negative rewards by the end of training only when the opponent was changing up to 4 steps per iteration.

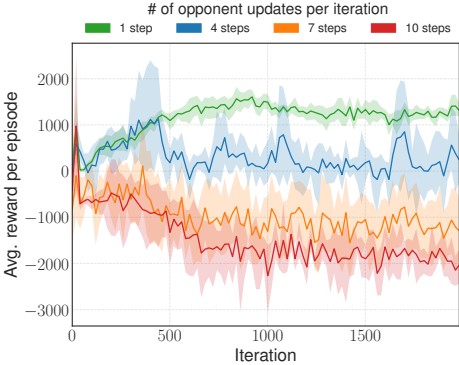

Fig. 11: Reward curves for a meta-learning agent trained against a learning opponent. Both agents were `Ants` with MLP policies. At each iteration, the opponent was updating its policy for a given number of steps using self-play, while the meta-learning agent attempted to learn to adapt to the distributional shifts. For each setting, the training process was repeated 15 times; shaded regions denote 90% confidence intervals.

