# OpenReview forum: "Continuous Adaptation via Meta-Learning in Nonstationary and Competitive Environments"
_ICLR.cc/2018/Conference — Accept (Oral)_

### Official Review · AnonReviewer3 · 2017-11-24

**Rating:** 8
**Confidence:** 4

**Review:**

This is a dense, rich, and impressive paper on rapid meta-learning. It is already highly polished, so I have mostly minor comments.

Related work: I think there is a distinction between continual and life-long learning, and I think that your proposed setup is a form of continual learning (see Ring ‘94/‘97). Given the proliferation of terminology for very related setups, I’d encourage you to reuse the old term.

Terminology: I find it confusing which bits are “meta” and which are not, and the paper could gain clarity by making this consistent. In particular, it would be good to explicitly name the “meta-loss” (currently the unnamed triple expectation in (3)). By definition, then, the “meta-gradient” is the gradient of the meta-loss -- and not the one in (2), which is the gradient of the regular loss.

Notation: there’s redundancy/inconsistency in the reward definition: pick either R_T or \bold{r}, not both, and maybe include R_T in the task tuple definition? It is also confusing that \mathcal{R} is a loss, not a reward (and is minimized) -- maybe use another symbol?

A question about the importance sampling correction: given that this spans multiple (long) trajectories, don’t the correction weights become really small in practice? Do you have some ballpark numbers?

Typos:
- “event their learning”
- “in such setting”
- “experience to for”

---

> ### Author Response · Authors · 2017-12-30
> **Thank you**
>
> Thank you for carefully reading the paper and the thoughtful comments. We answer the questions below:
>
> Related work:
> We agree that continuous adaptation is indeed a variation of continual learning. The updated version of the paper now points this out.
>
> Terminology:
> Thank you for suggestions. We have improved our terminology and notation throughout the paper, explicitly named the meta-loss, and renamed the inner loop gradient update (as given in Eq. 2) from “meta-update” to “adaptation update”.
>
> Notation:
> Initially, \mathcal{R}_T was standing for the risk (i.e., the expected loss, as commonly used in machine learning literature). This notation was indeed a bit confusing in the RL context where R is often used for rewards, so we have altered it.
>
> Importance sampling:
> Good point. Eq. 9 gives a general form of the estimator for \phi. In practice, the adaptation gradient (i.e., the gradient of L_{T_{i-1}} as now given in Eq. 9) decouples into a sum over time steps, so we compute importance weights for each time step (i.e., for each action) separately. The effective sample size in our experiments was no less than 20% of the given sample size. (Also, see our answer to a similar question asked by R2.)

---

### Official Review · AnonReviewer2 · 2017-11-28
**Review for Continuous Adaptation via Meta-Learning in Nonstationary and Competitive Environments**

**Rating:** 7
**Confidence:** 4

**Review:**

This paper proposed a gradient-based meta-learning approach for continuous adaptation in nonstationary and adversarial environment. The idea is to treat a nonstationary task as a sequence of stationary tasks and train agents to exploit the dependencies between consecutive tasks such that they can deal with nonstationarities at test time. The proposed method was evaluated based on a nonstationary locomotion and within a competitive multi agent setting. For the later, this paper specifically designed the RomoSumo environment and defined iterated adaptation games to test various aspect of adaptation strategies. The empirical results in both cases demonstrate the efficacy of the proposed meta-learned adaptation rules over the baselines in the few-short regime. The superiority of meta-learners is further justified on a population level.

The paper addressed a very important problem for general AI and it is well-written. Careful experiment designs, and thorough comparisons make the results conniving. I

Further comments:

1. In the experiment the trajectory number seems very small, I wonder if directly using importance weight as shown in (9) will cause high variance in the performance?

2. One of the assumption in this work is that trajectories from T_i contain some information about T_{i+1}, I wonder what will happen if the mutually information is very small between them (The extreme case is that two tasks are independent), will current method still perform well?

P7, For the RL^2 policy, the authors mentioned that “…with a given environment (or an opponent), reset the state once the latter changes” How does the agent know when an environment (or opponent) changes?

P10, “This suggests that it meta-learned a particular…” This sentence need to be rewritten.

P10, ELO is undefined

---

> ### Author Response · Authors · 2017-12-30
> **Thank you**
>
> Thank you for carefully reading the paper and the thoughtful comments. We answer the questions below:
>
> 1. Good point, the estimator in Eq. 9 may cause high variance in \phi. Three points to note:
> (i) We used importance weight correction only at execution time (indeed, the variance of the estimator hindered learning in such a regime; see footnote 2).
> (ii) Even though Eq. 9 shows the sum over K episodes, each episode consists of multiple time steps (typically 100-500 time steps, depending on the experiment) each of which is treated as a separate sample and gets an importance weight. Even with a limited number of episodes, we get quite a substantial number of time steps (with 3 episodes of 500 steps each, we get 1,500 time steps). In our experiments, the effective sample size was always reasonable (more than 20%), which worked at execution time (but not for learning).
> (iii) To compute adaptation updates using Eq. 9 in practice, meta-learners not only used the immediate past episode, but multiple previous episodes (see section 5.1, paragraph 2), which increased the number of samples and further helped to reduce the variance of the estimator.
>
> 2. No, the method is not designed to work in the regime with no mutual information between T_i and T_{i+1}. Our meta-learning approach targets to solve a zero-shot problem (i.e., do well at T_{i+1} without previous interaction experience with that particular task) knowing that tasks are sequentially dependent. If the tasks are independent, having some initial interaction with each new task is perhaps the only way to solve the problem.
>
> 3. In our setup, the number of episodes after which the environment/opponent changes is fixed. Moreover, we assume that the agent knows a priori the number of episodes or rounds after which the environment or opponent changes. This information is directly used by RL^2.

---

### Official Review · AnonReviewer1 · 2017-11-28
**This is a strong paper presenting a novel approach to few-shot eval-time adaptation in non-stationary environments, supported by multiple analyses in two synthetic domains.**

**Rating:** 9
**Confidence:** 2

**Review:**

---- Summary ----
This paper addresses the problem of learning to operate in non-stationary environments, represented as a Markov chain of distinct tasks. The goal is to meta-learn updates that are optimal with respect to transitions between pairs of tasks, allowing for few-shot execution time adaptation that does not degrade as the environment diverges ever further from the training time task set.

During learning, an inner loop iterates iterates over consecutive task pairs. For each pair, (T_i, T_{i+1}) trajectories sampled from T_i are used to construct a local policy that is then used to sample trajectories from T_{i+1}. By calculating the outer-loop policy gradient with respect to expectations of the trajectories sampled from T_i, and the trajectories sampled from T_{i+1} using the locally optimal inner-loop policy, the approach learns updates that are optimal with respect to the Markovian transitions between pairs of consecutive tasks.

The training time optimization algorithm requires multiple passes through a given sequence of tasks. Since this is not feasible at execution time, the trajectories calculated while solving task T_i are used to calculate updates for task T_{i+1} and these updates are importance weighted w.r.t the sampled trajectories' expectation under the final training-time policy.

The approach is evaluated on a pair of tasks. In the locomotion task, a six legged agent has to adapt to deal with an increasing inhibition to a pair of its legs. In the new RoboSumo task, agents have to adapt to effectively compete with increasingly competent components, that have been trained for longer periods of time via self-play.

It is clear that, in the locomotion task, the meta learning strategy maintains performance much more consistently than approaches that adapt through PPO-tracking, or implicitly by maintaining state in the RL^2 approach. This behaviour is less visible in the RoboSumo task (Fig 5.) but it does seem to present. Further experiments show that when the adaptation approaches are forced to fight against each other in 100 round iterated adaptation games, the meta learning strategy is dominant. However, the authors also do point out that this behaviour is highly dependent on the number of episodes allowed in each game, and when the agent can accumulate a large amount of evidence in a given environment the meta learning approach falls behind adaptation through tracking. The bias that allows the agent to learn effectively from few examples precludes it from effectively using many examples.

---- Questions for author ----
Updates are performed from \theta to \phi_{i+1} rather than from \phi_i to \phi_{i+1}. Footnote 2 states that this was due to empirical observations of instability but it also necessitates the importance weight correction during execution time. I would like to know how the authors expect the sample in Eqn 9 to behave in much longer running scenarios, when \pi_{\phi} starts to diverge drastically from \pi_{\theta} but very few trajectories are available.

The spider-spider results in Fig. 6 do not support the argument that meta learning is better than PPO tracking in the few-shot regime. Do you have any idea of why this is?

---- Nits ----
There is a slight muddiness of notation around the use of \tau in lines 7 & 9 in  of Algorithm 1. I think it should be edited to line up with the definition given in Eqn. 8.

The figures in this paper depend excessively and unnecessarily on color. They should be made more printer, and colorblind, friendly.

---- Conclusion ----
I think this paper would be a very worthy contribution to ICLR. Learning to adapt on the basis of few observations is an important prerequisite for real world agents, and this paper presents a reasonable approach backed up by a suite of informative evaluations. The quality of the writing is high, and the contributions are significant. However, this topic is very much outside of my realm of expertise and I am unfamiliar with the related work, so I am assigning my review a low confidence.

---

> ### Author Response · Authors · 2017-12-30
> **Thank you**
>
> Thank you for carefully reading the paper and thoughtful questions. We have improved the notation and the color-coding in the figures. We answer the questions below:
>
> - When \pi_{\phi} significantly diverges from \pi_{\theta}, the estimate given Eq. 9 would become of very high variance. In our setup, \pi_{\phi} was always at most a few gradient steps away from \pi_{\theta} in the parameter space. This gave difference in behaviors while keeping the effective sample size reasonable (always more than 20%). Much longer running scenarios may require a better estimator (i.e., of lower variance) which should also take into account the sequential structure of the tasks (e.g., a particle filter).
>
> - Good point, different methods yielded similar performance in the spider-spider experiments. This is because the agents tended to learn very similar behaviors regardless of the algorithm. The spectrum of behaviors learned by the agent highly depends on the morphology. From videos, we noticed that spiders always picked up a very particular fighting style, using front legs to kick the opponent and back legs to stabilize the posture, and never altered it during adaptation. This could be due to, perhaps, optimality of such behavior, but we did not further quantify this effect.

---

### Author Response · Authors · 2017-12-30
**Thank you**

We thank reviewers for their time and thoughtful feedback.

We have updated the submission: improved notation throughout the paper, resolved ambiguities, and improved color-coding in the plots. We answer specific questions raised in the reviews by separately replying to each of them.

---

### Public Comment · (anonymous) · 2018-02-18
**code**

Thanks for the work, do you plan to release the code recently?

---

### Public Comment · ~Yuchen_Lu1 · 2018-04-30
**Meta-learned Policy Starting Weak?**

Congratulations on winning the best paper, and I really enjoy reading the paper and the demo. However, I have some questions about this paper:

There is one thing that I notice in both experiments is that the policy trained from meta-learning starts with a weak point.  For example, at episode 1 in locomotion experiment,  meta-updates mostly perform the worst. I can guess there might be a trade-off between getting a good policy for a specific environment and a good policy for fast adaptation. Is this phenomenon also common in your experiments?

---

### Decision · Program_Chairs · 2018-01-29
**ICLR 2018 Conference Acceptance Decision**

**Decision:**

Accept (Oral)

**Comment:**

Looks like a great contribution to ICLR. Continuous adaptation in nonstationary (and competitive) environments is something that an intelligent agent acting in the real world would need to solve and this paper suggests that a meta-learning approach may be quite appropriate for this task.